# A Unified Fast Gradient Clipping Framework for DP-SGD

**Weiwei Kong**
Google Research
weiweikong@google.com

**Andres Muñoz Medina**
Google Research
ammedina@google.com

## Abstract

A well-known numerical bottleneck in the differentially-private stochastic gradient descent (DP-SGD) algorithm is the computation of the gradient norm for each example in a large input batch. When the loss function in DP-SGD consists of an intermediate linear operation, existing methods in the literature have proposed decompositions of gradients that are amenable to fast norm computations. In this paper, we present a framework that generalizes the above approach to arbitrary (possibly nonlinear) intermediate operations. Moreover, we show that for certain operations, such as fully-connected and embedding layer computations, further improvements to the runtime and storage costs of existing decompositions can be deduced using certain components of our framework. Finally, preliminary numerical experiments are given to demonstrate the substantial effects of the aforementioned improvements.

## 1 Introduction

Machine learning models — more specifically, neural network-based models — are becoming more popular in industrial applications, in user-facing products, and different scientific fields. The popularity of these models lies on their flexibility and ability to be trained on ever bigger datasets, which may contain personal information about individuals. As these models become bigger and more descriptive, ML practitioners need to ensure that the models, and their black-box interactions, do not reveal information about the data used to train the model. In fact, it has been shown repeatedly [18] that large neural-network models can be used to know if a particular example was used in the training data. Another line of attacks has demonstrated that one can actually reconstruct some training instances with simple interactions with a trained model [3].

The only known robust way of protecting against these attacks is to train models using differential privacy [10, 11]. Using this approach, ML practitioners provide an information theoretic guarantee that ensures that the final model does not depend on any individual example[1]. To date, the most popular method for training models with differential privacy is the differentially private stochastic gradient descent (DP-SGD) [1] method. In theory, the DP-SGD algorithm requires only minimal changes with respect to its non-private counterpart; one only requires to clip gradients observed in the training process and add some noise proportional to the clipping value. However, in practice, the (naïve) gradient clipping step has been shown to increase memory and computational costs in all popular learning platforms (JAX, Tensorflow, and Pytorch). More precisely, for a batch of size $n$ a naïve implementation of DP-SGD requires calculating $n$ gradients (one for each example in the batch) so they can be clipped. This is in stark contrast with most back-propagation-based training algorithms which calculate a single gradient. This implies a dependency on the runtime and memory in proportion to the batch size used to train the model, and for large models, this cost makes the

---

[1]In particular, it is impossible to distinguish whether or not an individual example is in the dataset, let alone reconstruct it.

37th Conference on Neural Information Processing Systems (NeurIPS 2023).

prospect of training with differential privacy simply not viable. Several papers [8, 17, 21] have also observed that large models need to be trained with large batch sizes to obtain competitive levels of utility, under reasonable amounts of privacy.

In [12], it was recognized that one could clip gradients without actually materializing every example gradient. This technique is referred to as *ghost clipping*. The initial proposal from [12] was specialized to fully connected, feed-forward neural networks that consist of only dense layers. The ghost clipping algorithm was later extended to handle convolution layers [13, 15] and attention layers [14].

However, each of the above solutions required an ad-hoc analysis on the correctness of their implementation. In this work we present a general analysis of the ghost clipping algorithm. To produce our results, we study this algorithm from the lens of linear operator theory. Our paper has three main contributions:

1. We unify years of ad-hoc analysis and interpretations of the ghost clipping algorithm under a single framework.

2. We provide a future-proof way of expanding the ghost-clipping technique to new layers with (possibly) nonlinear dependencies on their weights.

3. We demonstrate that by framing the ghost clipping problem in the language of linear operators, we can obtain a better performance of DP-SGD on models with embedding layers (crucial for personalization models consisting of embeddings of tens of millions of parameters) and fully connected layers with linear bias broadcasting operators.

Additionally, we show in our Appendix how to apply our framework to more complex tranforms/layers such as layer normalization and multi-head attention.

To complement the results of this paper, we open-sourced the general interface of the code using the TensorFlow Keras API[2]. By introducing an abstract interface, we also expect practitioners to easily extend the ghost clipping algorithm to any type of layer.

## 2 Notation and preliminaries

Throughout the paper, $(\mathcal{W}, \langle \cdot, \cdot \rangle)$ denotes a Hilbert space and $\| \cdot \|$ denotes its induced norm. Examples of $\mathcal{W}$ are $\mathbb{R}^d$ with the standard dot product and the space of matrices $\mathcal{W} = \mathbb{R}^{p \times q}$ with the Euclidean (Frobenius) inner product between $A, B \in \mathcal{W}$ given by $\langle A, B \rangle = \text{tr}(A^\top B)$. For two matrics $A, B$ we let $\|A\|$ denote the Frobenius norm of $A$.

Given two Hilbert spaces $(\mathcal{W}, \langle \cdot, \cdot \rangle_\mathcal{W})$, $(\mathcal{Y}, \langle \cdot, \cdot \rangle_\mathcal{Y})$, we denote linear operators between them by italicized letters ($\mathcal{A} \colon \mathcal{W} \to \mathcal{Y}$) and $\mathcal{A}^* \colon \mathcal{Y} \to \mathcal{W}$ to be the *adjoint* of $\mathcal{A}$. That is, $\mathcal{A}^*$ is the unique linear operator that satisfies

$$\langle y, \mathcal{A}w \rangle_\mathcal{Y} = \langle \mathcal{A}^* y, w \rangle_\mathcal{W} \quad \forall w \in \mathcal{W}, \quad \forall y \in \mathcal{Y}. \tag{1}$$

Let $(\mathcal{W}, \langle \cdot, \cdot \rangle_\mathcal{W})$ and $(\mathcal{Y}, \langle \cdot, \cdot \rangle_\mathcal{Y})$ denote two Hilbert spaces with respective induced norms $\| \cdot \|_\mathcal{W}$ and $\| \cdot \|_\mathcal{Y}$. Moreover, let $\psi \colon \mathcal{W} \to \mathcal{Y}$ be an arbitrary function. The *Fréchet derivative* of $\psi$ at $w_0 \in \mathcal{W}$ is given by the unique bounded linear operator $D\psi(w_0) \colon \mathcal{W} \to \mathcal{Y}$ satisfying

$$\lim_{\delta \to 0} \frac{\|\psi(w_0 + \delta) - \psi(w_0) - D\psi(w_0)\delta\|_\mathcal{Y}}{\|\delta\|_\mathcal{W}} = 0.$$

We say $\psi$ is differentiable if its Fréchet derivative exists for all $w_0 \in \mathcal{W}$. Throughout this paper we will use two special properties of the Fréchet derivative: the chain rule and the existence of gradients. Let $(\mathcal{Z}, \langle \cdot, \cdot \rangle_\mathcal{Z})$ be another Hilbert space and $\phi \colon \mathcal{Y} \to \mathcal{Z}$ be given. The *chain rule* provides us with a simple way to calculate the derivative of the function $\phi \circ \psi \colon \mathcal{W} \to \mathcal{Z}$, namely,

$$D(\phi \circ \psi)(w_0) = D\phi(\psi(w_0))D\psi(w_0).$$

The Fréchet derivative of $\psi$ at $w_0$ with respect to a subset of variables $u$ is denoted by $D_u\phi(w_0)$. Finally, $\nabla\psi(w_0) \in \mathcal{W}$ denotes the (unique) gradient of a function $\psi$ at $w_0$, which satisfies

$$D\psi(w_0)\delta = \langle \nabla\psi(w_0), \delta \rangle_\mathcal{W} \quad \forall \delta \in \mathcal{W} \tag{2}$$

The existence of the gradient is guaranteed by the well-known Riesz-Fréchet Representation Theorem [16]. The gradient of $\psi$ at $w_0$ with respect to a set of variables $u$ is denoted by $\nabla_u\psi(w_0)$.

---

---

**Algorithm 1** DP-SGD algorithm

---

**Input:** *Data sample $S$, parameter initialization $w_0$, number of rounds $T$, learning rate $\eta$, noise multiplier $\sigma$, clipping norm $C$.*

Set $w = w_0$

**for** $t = 1, \ldots, T$ **do**

    Get a batch of inputs $B \subset \mathcal{S}$

    Calculate per example gradients $g_x = \nabla_w \ell(x, w)$ for every $x \in B$

    Clip gradients $\widetilde{g}_x = \min\left\{ \frac{C}{\|g_x\|}, 1 \right\} g_x$

    Calculate private batch gradient $\widetilde{g} = \frac{1}{|B|} \sum_{x \in B} \widetilde{g}_x + N\left(0, \frac{C^2 \sigma^2}{|B|^2} I\right)$

    Update model $w = w - \eta \widetilde{g}$

**end for**

**return** $w$

---

**Private Stochastic Gradient Descent** We now turn our attention to the DP-SGD algorithm and specifically, to its instantiation for neural network-based models. Let $\mathcal{X}$ be an arbitrary space and $\mathcal{S} = \{x_1, \ldots, x_n\} \subset \mathcal{X}$ be a sample of examples. Let $\mathcal{W}$ denote a parameter space and $h \colon \mathcal{X} \times \mathcal{W} \to \mathbb{R}$ denote an arbitrary function. The stochastic gradient descent (SGD) algorithm solves the optimization problem

$$\min_{w \in \mathcal{W}} \frac{1}{n} \sum_{i=1}^{n} h(x_i, w)$$

by iteratively updating the model paramters using gradients of the loss over a batch $B \subset \mathcal{S}$ of data.

The celebrated DP-SGD algorithm was introduced by [1] as a simple modification on the SGD algorithm to make it private[3]. Specifically, the DP-SGD algorithm, given in Algorithm 1, is identical to the SGD algorithm except in two steps:

1. The DP-SGD algorithm needs to calculate $|B|$ (so-called) per-example gradients in order to clip them to have a bounded norm.

2. Adding Gaussian noise $N(0, C^2 \sigma^2 / |B|^2)$ for some noise multiplier $\sigma > 0$.

Notice that the first step has a prohibitively large cost for large networks. Specifically, in the per example gradient calculation, notice that that the computational and memory usage of a naïve implementation[4] of DP-SGD algorithm increases as $\Theta(n_w |B|)$, where $n_w$ is the number of parameters in the network. This increase in resources effectively negates the advantages of the backpropagation algorithm. In this paper we show how to run the DP-SGD algorithm with only a small constant increment in both the memory footprint and runtime of traditional SGD.

We will focus on the scenario where the learner is trying to minimize the loss across the space of neural networks. That is, the parameter vector $w$ is a concatenation of $k$ parameters $(w_1, \ldots, w_k)$ and there exists $k$ functions $\phi_1, \ldots, \phi_k$, and a loss function $\ell$ such that

$$h(x, w) = \ell(x, \phi_k(w_k, \phi_{k-1}(w_{k-1}, ..., \phi_1(w_1, x)))).$$

Note that $\{\phi_j\}$ correspond to the layers of the network and $\{w_j\}$ are the vectors parameterizing these layers.

## 3 Previous work

The DP-SGD algorithm was first introduced by [1]. Due to its simplicity to adapt to standard machine learning frameworks it is now probably the most popular method for training private

---

[3]When talking about privacy we are referring to the concept of differential privacy. We refer the reader to [11] for a comprehensive guide on differential privacy. In this work we focus on the computational aspects and not the privacy properties of the DP-SGD algorithm

[4]Alternatively, one could consider sequentially computing the gradient norms to remove the linear memory scaling in $|B|$. However, this approach may prohibitively increase the runtime cost in practice.

machine learning models. In the past few years, a large number of papers have been devoted to improving the privacy-utility trade-offs of DP-SGD [4] as well as demonstrating that DP-SGD can be used on multiple tasks [7, 9, 2]. While the main focus on this line of research has been in understanding the privacy-utility trade-offs, some authors did touch on the performance issues of calculating per-example gradients. A simple solution to reduce the memory and computational blow-up of the DP-SGD algorithm is to calculate the gradient with respect to a micro-batch of examples instead. While this reduced the computational cost of running DP-SGD, it came at the cost of decreased quality of the model learned. Some authors [2] got around this performance issues by using some JIT compilation features of JAX. However these authors still observed an increase in the memory use of their private implementations.

In order to speed up the gradient clipping step without sacrificing quality, [12] proposed the ghost clipping technique (see Section 4). ITs general idea is that one can obtain a private gradient estimate $\widetilde{g}$ without materializing each per example gradient as long as one knows the norm of each per-example gradient. Moreover, the authors show that one can easily calculate these norms by using information already materialized in the forward and backward passes of the back propagation algorithm for training neural networks. The results however were limited to neural networks consisting of only dense layers. The ghost clipping trick was later on applied to train transformers [14] and image classification models which consist of networks with convolution layers [15].

The above works provide ad-hoc proofs that their implementation of the ghost clipping algorithm is correct for their particular layer implementation but do not attempt to extend their results to arbitrary layers. In this work we present a way to extend the ghost clipping techinque to arbitrary layers and pinpoint what properties of each layer are crucial for enabling an efficient implementation of the DP-SGD algorithm.

Finally, it is also worth mentioning that paper [6] presents an efficient implementation of the ghost clipping technique in PyTorch for models with fully-connected, embedding, convolution, and normalization layers using variable caching to avoid recomputing certain intermediate parameter gradients. Our implementation also takes advantage of variable caching, but does this implicitly through TensorFlow's `GradientTape` API[5].

## 4    The Ghost Clipping Algorithm

Algorithm 1 suggests that in order to implement DP-SGD one must calculate every per-example gradient, clip it to achieve a bounded norm and then aggregate it to obtain the private gradient estimate $\widetilde{g}$. The main observation from [12] was that, with knowledge of the norms $\|g\|_x$ one could estimate $\widetilde{g}$ without explicitly materializing each per example gradient. To do this, the authors define weights $r_x = \min\left\{C/\|g_x\|, 1\right\}$ and a new weighted sum

$$S(w) = \sum_{x \in B} r_x h(x, w).$$

Treating the weights $r_x$ as fixed (with respect to the parameters $w$), one can use the linearity of the gradient operator to see that $\nabla S(w) = \widetilde{g}$. Crucially, for neural networks, the gradient of $S$ can be efficiently calculated using a single back-propagation step. The second contribution of [12] was showing that, for neural networks consisting of dense layers, one can also efficiently calculate $\|g_x\|$ for every $x \in B$ using only a single forward and backward pass of the back propagation algorithm. That is, the cost of calculating a private gradient is simply twice that of calculating a non-private gradient step.

In the subsections below, we present a general formulation for calculating $\|g_x\|$ in a single forward and backward pass for arbitrary feed-forward neural networks, under the assumption that:

  ▷ $\ell(x, \cdot), \phi_1(\cdot, x), \ldots, \phi_k(\cdot, x)$ are Fréchet differentiable for every $x \in \mathcal{X}$;

  ▷ each function $\phi_i(\cdot, x)$ can be decomposed as the composition of at least two Fréchet differentiable subfunctions.

---

[5]See https://www.tensorflow.org/api_docs/python/tf/GradientTape

---

**Algorithm 2** General Gradient Norm Framework

---

**Input:** *Data* $x \in \mathcal{X}$, *parameter* $w = (w_1, \ldots, w_k)$, *and functions* $\{\psi_{x,i}\}_{i=1}^k$, $\{Z_{x,i}\}_{i=1}^k$ *satisfying*

$$h(x, \tilde{w}) = \ell_x \circ \psi_{x,i} \circ Z_{x,i}(\tilde{w}_i), \quad i = 1, \ldots, k \tag{5}$$

*for any* $\tilde{w} = (\tilde{w}_1, \ldots, \tilde{w}_k) \in \mathcal{W}$, *where* $\ell_x(\cdot) \equiv \ell(x, \cdot)$.

**for** $i = 1, \ldots, k$ **do**
    Compute $\zeta_{x,i} := Z_{x,i}(w_i)$
    Store a "nice" representation $\Omega_{x,i}(\cdot)$ of the squared-norm function $g \mapsto \|[DZ_{x,i}(w_i)]^*(g)\|^2$
    for any gradient $g$ of the function $\ell_x \circ \psi_{x,i}(\cdot)$
**end for**
**for** $i = 1, \ldots, k$ **do**
    Compute $g_{x,i} := \nabla_{\zeta_{i,x}}(\ell_x \circ \psi_{x,i})(\zeta_{x,i})$ and $\tau_{x,i} := \Omega_{x,i}(g_{x,i})$
**end for**
**return** $(\sum_{i=1}^k \tau_{x,i})^{1/2}$

---

## 4.1 The General Norm Calculation Algorithm

Let $w = (w_1, \ldots, w_k)$ parameterize a neural network. We first note that $\|\nabla_w h(x,w)\|^2 = \sum_{j=1}^k \|\nabla_{w_k} h(x,w)\|^2$. Hence, we can focus on efficiently calculating the norm of the gradient corresponding to the parameters of each layer. Let us then fix a layer and denote $\bar{w} \in \bar{\mathcal{W}}$ to be its parameter vector for some restricted space $\bar{\mathcal{W}}$. Moreover, let the input $x$ be fixed. We see that, as a function of the layer parameters $\bar{w}$ only, the loss $h$ has the form

$$h(x, \bar{w}) = \ell_x \circ \phi_x(\bar{w}) \quad \forall x \in \mathcal{X}, \tag{3}$$

where $\phi_x$ corresponds to operations performed by the fixed layer and all subsequent layers in the network, and $\ell_x(\cdot) \equiv \ell(x, \cdot)$ corresponds to the loss function, e.g., mean-squared error, for the input example $x$. The following result, whose proof can be found in the Appendix, gives an expression for the gradient of $h$ under any decomposition of $\phi_x$.

**Proposition 4.1.** *Let* $x \in \mathcal{X}$ *be fixed and let* $(\ell_x, \phi_x)$ *be a pair of Fréchet differentiable functions satisfying* (3). *Moreover, let* $(\psi_x, Z_x)$ *be a pair of Fréchet differentiable functions satisfying* $\phi_x = \psi_x \circ Z_x$, *and denote*

$$\mathcal{A} = \mathcal{A}_x(\bar{w}) := DZ_x(\bar{w}), \quad g = g_x(\bar{w}) := \nabla(\ell_x \circ \psi_x)(Z_x(\bar{w})). \tag{4}$$

*Then,* $\nabla_{\bar{w}} h(x, \bar{w}) = \mathcal{A}^* g$.

The above result gives us an alternative way of computing $\|\nabla_{\bar{w}} h(x, \bar{w})\|^2$, namely, (i) pick a decomposition $(\psi_x, Z_x)$ of $\phi_x$ and (ii) compute $\|\mathcal{A}^* g\|^2$ in some efficient manner. In later subsections, we provide examples of layers and decompositions where this two-step approach is drastically more efficient than the naive approach of materializing $\nabla_{\bar{w}} h(x, \bar{w})$ and computing its Euclidean norm.

In view of the decomposition in Proposition 4.1, we present a unified method for computing $\|\nabla h(x, w)\|$ in Algorithm 2. Specifically, it consists of two loops over the parameters $w_i$. The initial loop is performed to obtain the intermediate outputs $\zeta_{i,x}$ and some sort of "nice" or efficient representation of the squared-norm functions $\Omega_{i,x}$. The follow-up loop, also over the parameters $w_i$, computes some necessary intermediate gradients and combines them with the functions $\Omega_{i,x}$ to obtain the gradient decomposition (and subsequent norm computation) given by Proposition 4.1.

## 4.2 Efficient Implementation of Algorithm 2

We now describe how Algorithm 2 can be implemented efficiently when $h(\cdot, \cdot)$ is formed by a neural network $\mathcal{N}$. First, it is shown in Appendix B.1 that the intermediate quantities $\{\zeta_{i,x}\}_{i=1}^k$ and $\{g_{i,x}\}_{i=1}^k$ can be computed in a single forward and backward pass of $\mathcal{N}$ (as opposed to naïvely traversing the network $\Theta(k)$ times). Second, it is shown in Appendix B.2 that for any $i$, the batch gradients $\{\nabla_{\zeta_{x,i}}(\ell_x \circ \psi_{x,i})(\zeta_{x,i})\}_{x \in B}$ can be obtained a single (batched) backward pass of $\mathcal{N}$ if (i) condition (5) holds for every $x \in B$ and (ii) the gradient $g_{x,i}$ is replaced by $\nabla_{\zeta_{x,i}}\{\sum_{x \in B} \ell_x \circ \psi_{x,i}(\zeta_{x,i})\}$. Consequently, if we ignore the costs of forming and evaluating $\Omega_{x,i}(\cdot)$ for each $x \in B$

and $i = 1, \ldots, k$, then the above facts imply that we can obtain the norms $\{\|\nabla h(x, w)\|\}_{x \in B}$ using only one additional forward and (batched) backward pass of $\mathcal{N}$.

# 5 Efficient Squared-Norm Functions

This section presents efficient representations of the functions $\Omega_i(\cdot)$ in Algorithm 2 for some basic layer functions $\phi_x(w) \equiv \phi(w, x)$. Examples involving more complicated layer functions (e.g., layer normalization and multi-head attention) can be found in the Appendix.

Each subsection begins with a precise description of $\phi_x(\cdot)$, presents a decomposition $\phi(\cdot) = \psi_x \circ Z_x(\cdot)$, and gives an analysis of the runtime and storage costs of an efficient representation of the function $\Omega_x(\cdot) \equiv \|[DZ_x(w)]^*(\cdot)\|^2$. For the sake of conciseness, the proofs of these decompositions are given in the Appendix.

In addition, we make comparisons of our approach with the naive approach (of computing gradients for each example) and other ghost clipping-like approaches. The results of these comparisons are summarized in Tables 1–2 (see the first paragraphs in Subsections 5.1–5.3 for descriptions of the variables/dimensions).

|  | Naive | Ghost Clipping | **Ours** |
|---|---|---|---|
| Fully-Connected[6] | $\Theta(|B|\{pq + m\})$ | $\Theta(r^2 + |B|\{rq\}^2)$ | $\Theta(|B|r^2)$ |
| Embedding[7] | $\Theta(|B|qd)$ | $O(|B|q^2)$ | $\Theta(|B|\tilde{q})$ |
| Rank-$k$ Approx. | $\Theta(|B|nk)$ | - | $O(1)$ |

Table 1: Asymptotic storage costs for computing $\{\|\nabla h(x, W)\|\}_{x \in B}$. For ghost clipping and our approach, this is the storage cost of representing the squared-norm function $\Omega_x(\cdot)$ on the entire batch $B$.

|  | Naive | Ghost Clipping | **Ours** |
|---|---|---|---|
| Fully-Connected[6] | $\Theta(|B|\{pq + m\})$ | $\Theta(r^2\{p + |B|q^2\} + m\{rq\}^2)$ | $\Theta(r^2\{p + |B|q\})$ |
| Embedding[7] | $\Theta(|B|qd)$ | $O(|B|q^2d)$ | $\Theta(|B|\{q \log \tilde{q} + \tilde{q}d\})$ |
| Rank-$k$ Approx. | $\Theta(|B|n^2k)$ | - | $\Theta(n^2k + |B|)$ |

Table 2: Asymptotic runtime costs for computing $\{\|\nabla h(x, W)\|\}_{x \in B}$. For ghost clipping and our approach, this includes the time used to generate, represent, and evaluate the squared-norm function $\Omega_x(\cdot)$ on the entire batch $B$.

## 5.1 Fully-Connected Layers

Given variables $V \in \mathbb{R}^{p \times q}$ and $b \in \mathbb{R}^m$, a layer input $U_x \in \mathbb{R}^{r \times p}$, an activation function $\alpha : \mathbb{R}^{r \times q} \mapsto \mathbb{R}^{r \times q}$, and a linear broadcasting operator $\mathcal{Q} : \mathbb{R}^m \mapsto \mathbb{R}^{r \times q}$ satisfying $m \mid rq$, the standard fully-connected layer function $\phi_x(\cdot)$ is given by

$$\phi_x(V, b) = \alpha(U_x V + \mathcal{Q}b). \tag{6}$$

Typically, $p$, $q$, $r$, and $m$ are called the *input dimension*, *output dimension*, *channel dimension*, and *bias dimension*, respectively. Usually, it is the case that $r \ll \min\{p, q\}$ and a common case is $r = 1$.

We now consider the squared-norm function $\Omega_x : \mathbb{R}^{r \times q} \mapsto \mathbb{R}$ generated by the choice of $Z_x(V, b) := U_x V + \mathcal{Q}b$. Denoting $\mathcal{A}$ as in (4), for some $\bar{w} = (V, b)$, we have that

$$\Omega_x(g) = \|\mathcal{A}^* g\|^2 = \|U_x^* g\|^2 + \|\mathcal{Q}^* g\|^2 = \langle U_x U_x^*, gg^* \rangle + \|\mathcal{Q}^* g\|^2, \tag{7}$$

for any $g \in \mathbb{R}^{r \times q}$. Hence, $\Omega_x(\cdot)$ can be efficiently represented by $U_x U_x^*$ and an efficient representation of the function $g \mapsto \|\mathcal{Q}^* g\|^2$. For an example of the latter, suppose $\mathcal{Q}$ is the operator that repeats $b$ in an $r$-by-$q$ matrix row-wise $\rho := rq/m$ times. Then, defining the row/column maps

$$\pi(\ell, k) := 1 + \left\lfloor \frac{(\ell - 1)m + (k - 1)}{q} \right\rfloor, \quad \xi(\ell, k) := 1 + \{[(\ell - 1)m + (k - 1)] \mod r\},$$

---

[6]Based on the example in Section 5.
[7]Using Algorithm 3 for ours.

it follows from the definition of the adjoint that

$$[\mathcal{Q}^* g]_k = \sum_{\ell=1}^{\rho} g_{\pi(\ell,k),\xi(\ell,k)} \quad \forall g \in \mathbb{R}^{r \times q}, \quad k = 1, \ldots, m.$$

For this example, we incur a compute (resp. storage) cost of $\Theta(r^2 p)$ (resp. $\Theta(r^2)$) for materializing $U_x U_x^*$ and a compute and storage cost of $O(1)$ for the function $g \mapsto \|\mathcal{Q}^* g\|^2$, which does not depend on the data $x$. On the other hand, evaluation of $\Omega_x(\cdot)$ incurs: (i) a compute (resp. storage) cost of $\Theta(r^2 q|B|)$ (resp. $\Theta(r^2|B|)$) for materializing each $gg^*$ in the batch $B$ and then computing the inner product term $\langle U_x U_x^*, gg^* \rangle$ and (ii) a compute cost of $\Theta(rq|B|)$ for evaluating $\|\mathcal{Q}^* g\|$, which only requires a single pass through the entries of $g$ for each example. Consequently, the total runtime (resp. storage) cost for a batch of examples $B$ is

$$\Theta(r^2 p + r^2 q|B| + rq|B|) = \Theta(r^2\{p + |B|q\})$$
$$[\text{resp. } \Theta(1 + r^2 + r^2|B|) = \Theta(|B|r^2)].$$

Let us now make a comparison of this decomposition with the naive and ghost clipping approaches. For our discussion, suppose $\mathcal{Q}$ is as in the previous example.

In the naive approach of computing $\nabla_{(V,b)} h(x, (V,b))$ for each $x \in B$, it is straightforward[8] to show that the compute (resp. storage) costs are $\Theta(|B|\{pq + m\})$ (resp. $\Theta(|B|\{p + r\}q)$).

For the classic ghost clipping approach considered in [12, 13], our approach is identical in the case of $\mathcal{Q} \equiv 0$. When $\mathcal{Q} \not\equiv 0$, this approach instead groups $\mathcal{Q}$ together with $U_x$ and suggests the decomposition in (7) but with $U_x = [U_x; \mathcal{Q}]$ and $\mathcal{Q} = 0$. Let us now analyze the costs in the worst-case scenario when $\mathcal{Q}\mathcal{Q}^*$ is dense, e.g., when $m = 1$. In order to evaluate $\langle \mathcal{Q}\mathcal{Q}^*, gg^* \rangle$ using a consistent representation of $\mathcal{Q}\mathcal{Q}^*$, we must materialize $\mathcal{Q}$ (resp. $g$) as a $rq$-by-$m$ matrix ($rq$-by-1 column vector) and compute $\mathcal{Q}\mathcal{Q}^* \in \mathbb{R}^{rq \times rq}$ (resp. $gg^* \in \mathbb{R}^{rq \times rq}$) using matrix multiplication. Clearly, the compute (resp. storage) costs for $\langle \mathcal{Q}\mathcal{Q}^*, gg^* \rangle$ are then $\Theta(m\{rq\}^2 + |B|\{rq\}^2)$ (resp. $\Theta(mrq + |B|\{rq\}^2)$) given $\mathcal{Q}$ which, together with the costs for $\|U_x^* g\|^2$, yield the complexity in Table 2 (resp. Table 1).

## 5.2 Embedding Layers

Given variables $W \in \mathbb{R}^{r \times d}$, input indices $\pi(x) = [\pi_1(x), \ldots, \pi_q(x)] \in \{1, \ldots, r\}^q$, the standard embedding layer function $\phi_x(\cdot)$ is given by

$$\phi_x(W) = Y_{\pi(x)} W,$$

where $Y_{\pi(x)} \in \mathbb{R}^{q \times r}$ is the one-hot matrix whose $\ell$-th row is the $\pi_\ell(x)$-th basis vector in $\mathbb{R}^r$. Typically, $r$, $d$, and $q$ are called the *vocabulary size*, *embedding dimension*, and *number of queries*, respectively. Usually, it is the case that $\max\{d, q\} \ll r$, and a common case is $q = 1$.

We now consider the squared-norm function $\Omega_x : \mathbb{R}^{q \times d} \mapsto \mathbb{R}$ generated by the choice of $Z_x(W) = \phi_x = Y_{\pi(x)} W$ for any gradient $g = \nabla(\ell_x \circ \psi_x)(Z_x(W))$. First, define the quantities

$$n_k(x) := |\{i : \pi_i(x) = k, i = 1, \ldots, q\}|, \quad \tilde{q} := |\{i : n_i(x) > 0\}|,$$

for $k = 1, \ldots, q$, where $n_k(x)$ denotes the number times index $k$ appears in $\pi(x)$. Next, notice that $g$ contains at most $\tilde{q}$ unique rows, due to the definition of the gradient and the fact that $Z_x(W)$ has at most $\tilde{q}$ unique rows. Denoting $\mathcal{A}$ as in (4), for some $\bar{w} = W$, it can be shown that

$$\Omega_x(g) = \|\mathcal{A}^* g\|^2 = \|Y_{\pi(x)}^* g\|^2 = \sum_{i=1}^{r} n_i^2(x) \cdot \|\tilde{g}_i\|^2,$$

where $\tilde{g}_i$ denotes any row $j$ of $g$ where $\pi_j(x) = i$ (note that all rows of $g$ that satisfy this condition are the same). Hence, $\Omega_x(\cdot)$ can be efficiently represented by the nonzero values of $\{n_k(x)\}_{k=1}^r$. An efficient algorithm using $O(\tilde{q} \log q)$ runtime and $O(\tilde{q})$ extra storage, for computing $\{n_x(x)\}_{k=1}^r$, for a fixed $x \in B$, is given in Algorithm 3 of the Appendix. On the other hand, the evaluation of $\Omega_x(\cdot)$ only requires evaluating the $\tilde{q}$ unique rows of an input $g \in \mathbb{R}^{q \times d}$ (given by $\pi(x)$) and multiplying

---

[8]Hint: the cost of storing the gradients for a single batch is equal to the cost of storing the weights.

them with the $\tilde{q}$ nonzero cached values $\{n_k(x)\}_{k=1}^r$, which can be done with a $\Theta(|B|\tilde{q}d)$ runtime cost for a batch $B$. Consequently, the total runtime (resp. storage) cost for a batch of examples $B$ is $\Theta(|B|\{q\log\tilde{q} + \tilde{q}d\})$ (resp. $\Theta(|B|\tilde{q})$) using Algorithm 3.

Let us now make a comparison of this decomposition with the naive approach and the ghost clipping approach in [13].

In the naive approach of computing $\nabla_W h(x, W)$ for each $x \in B$, it is straightforward[9] to show that the compute (resp. storage) costs are $\Theta(|B|qd)$ (resp. $\Theta(|B|qd)$).

In the classic ghost clipping approach, we treat $\phi_x(\cdot)$ as a linear operator, compute $Y_{\pi(x)}Y_{\pi(x)}^*$, and choose the representation $\Omega_x(g) = \langle Y_{\pi(x)}Y_{\pi(x)}^*, gg^* \rangle$. In the worst-case scenario, it is straightforward to see that $Y_{\pi(x)}Y_{\pi(x)}^*$ can be fully dense, e.g., $\tilde{q} = 1$. In this setting, even if $Y_{\pi(x)}$ is sparsely represented by a small set of $\tilde{q}$ unique indices, computing $Y_{\pi(x)}Y_{\pi(x)}^*$ still incurs a compute and storage cost of $O(q^2)$. Now, since each example gradient $g$ consists of $q$ embedding vectors in $\mathbb{R}^d$, for a batch $B$, the compute (resp. storage) cost of materializing $gg^*$ is $\Theta(|B|q^2d)$ (resp. $\Theta(|B|q^2)$). Combining the above complexities with the $\Theta(|B|q^2)$ runtime cost of computing the desired inner products, i.e. (7) with $U_x = Y_{\pi(x)}$ for $x \in B$, we obtain the complexities in Tables 1–2.

### 5.3 Low Rank Approximation Layer

Given input matrix $U_x \in \mathbb{R}^{n \times n}$, one way [19, 20] to encourage a rank-$k$ (or lower) approximation of $U_x$ is to add the intermediate layer transform

$$\phi_x(V) = \|U_x - VV^*\|^2 + \rho \circ \sigma(VV^*)$$

for some $V \in \mathbb{R}^{n \times k}$, where $\rho(\cdot)$ is a sparsity promoting regularizer (e.g., $\ell_1$ norm, SCAD, MCP) and $\sigma(\cdot)$ is the function that maps matrices to their singular values. Expanding the norm term, note that the above function can be equivalently (ignoring terms depending solely on $U_x$) expressed as

$$\phi_x(V) = -2\langle U_x, VV^* \rangle + \mathcal{R}(V), \tag{8}$$

for some function $\mathcal{R} : \mathbb{R}^{n \times k} \mapsto \mathbb{R}$. Note that $\mathcal{R}(V)$ does not depend on $x$ and, hence, its computation does not depend the batch $B$.

In view of (8), we now consider the squared-norm function $\Omega_x : \mathbb{R} \mapsto \mathbb{R}$ generated by the choice of $Z_x(V) = \langle U_x, VV^* \rangle / 2$, where, clearly, one has $\phi_x(V) = -4Z_x(V) + \mathcal{R}(V)$. Denoting $\mathcal{A}$ as in (4), for some $\bar{w} = V$, it can be shown that

$$\Omega_x(g) = \|\mathcal{A}^*g\|^2 = \frac{g^2}{4}\|(U_x + U_x^*)V\|^2,$$

for $g \in \mathbb{R}$. Hence, $\Omega_x(\cdot)$ can be efficiently represented by the scalar $\|(U_x + U_x^*)V\|^2$. It is straightforward to see that computing $\|(U_x + U_x^*)V\|$ requires only a $\Theta(n^2k)$ runtime cost and a $\Theta(1)$ storage cost. Moreover, evaluation of $\Omega_x(\cdot)$, given $\|(U_x + U_x^*)V\|$, requires only a $O(1)$ runtime cost.

In the naive approach of computing $\nabla_V h(x, V)$ for each $x \in B$, it is straightforward to see that the compute (resp. storage) costs is $\Theta(|B|n^2k)$ (resp. $\Theta(|B|nk)$) due to the excessive computation (resp. storage) of $g(U_x + U_x^*)V$ for $x \in B$. The authors are not aware of any ghost clipping-like techniques in the nonlinear setting.

## 6 Numerical Experiments

This section presents numerical experiments that compare our proposed adjoint-based framework (`Adjoint`) against the naïve implementation of DP-SGD (`Naive`), which computes gradients for each example in a batch, and the classic ghost clipping frameworks (`GhostClip`) that are described in Subsections 5.1 and 5.2. Specifically, it presents runtimes and memory costs for the gradient norm computation of fully-connected and embedding layers.

---

[9]Hint: we need to store (and take the norm of) $q$ embedding vectors in $\mathbb{R}^d$ for each example in the batch $B$.

Each problem instances was run on a cloud computing platform consisting of (i) 112 Intel(R) Xeon(R) Platinum processors with 28 cores each, (ii) 64 GB of RAM, (iii) Python 3.10.11, and (iv) Tensorflow 2.14. For simplicity, memory usage was measured as the peak amount of heap memory utilized in the run of single gradient norm computation. We also simplify our computations by utilizing batches of size $|B| = 1$ for the first two subsections. The loss function used in our experiment, $\ell_x(\cdot)$, is the mean-squared-error. To reduce the variance of the results in the first two subsections, we repeat each problem instance 20 times and report only the median runtime and memory cost over the repetitions.

## 6.1 Fully-Connected Layer

Figure 1 presents numerical results for the setting considered in Subsection 5.1, where $\mathcal{Q}$ is the (linear) broadcasting operator that duplicates the bias $rq/m$ times to match the dimension of the layer outputs. It specifically plots the effect of the bias dimension $m$ for various values of the output dimension $q$. For simplicity, all problem instances fix an input and channel dimension of 2 and 4096, respectively. Additional experiments, involving the effect of batch size, are given in Appendix F.

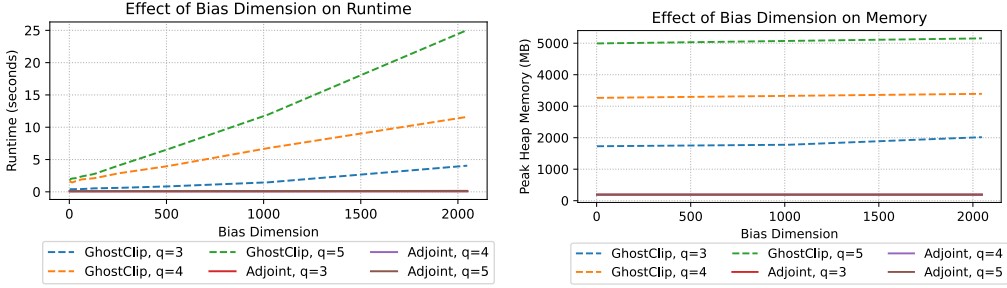

Figure 1: Runtime and memory cost graphs for fully-connected layer computations with bias dimensions $m = \{2^1, 2^2, \dots, 2^{11}\}$ and output dimensions $q = 3, 4, 5$.

The results in Figure 1 demonstrate that the runtime and memory costs of `Adjoint` are marginal compared to those of `GhostClip` for fully-connected layers. These results also support the analysis of Subsection 5.1 in that: (i) `GhostClip`'s runtime has a stronger positive dependence on the bias dimension $m$ than `Adjoint`'s runtime and (iii) `GhostClip`'s runtime has a strong positive dependence on the output dimension $q$.

## 6.2 Embedding Layer

Figure 2 presents numerical results for the setting considered in Subsection 5.2. It specifically plots the effect of the number of queries $q$ for various values of the vocabulary size $r$. For simplicity, all problem instances fix the embedding dimension to be 10 and the embedding indices $\{\pi_i(x)\}_{i=1}^q$ are chosen uniformly at randomly from the set $\{1, \dots, r\}$.

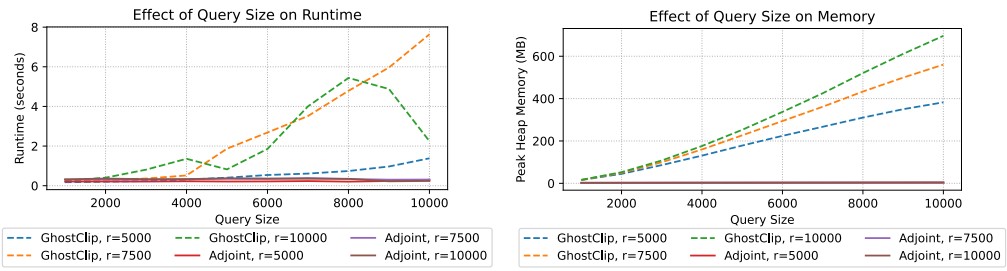

Figure 2: Runtime and memory cost graphs for embedding computations with query sizes $q = \{1000, 2000, \dots, 10000\}$ and vocabulary sizes $r = 5000, 7500, 10000$.

The results in Figure 2 demonstrate that the runtime and memory costs of `Adjoint` are marginal compared to those of `GhostClip` for embedding layers. Moreover, the memory cost graph supports

the analysis in Subsection 5.2 in that `GhostClip`'s memory cost has a significantly stronger positive dependence on the query size $q$ compared to `Adjoint`'s memory cost.

## 6.3 Small BERT model

Figure 3 presents numerical results for an end-to-end training run of a small BERT model that consists of dense, embedding, normalization, and multi-head attention sublayers. It specifically plots the effect of the batch size $|B|$ on the runtime and peak memory usage of training the model. To be more precise, the BERT model is an instance of the TensorFlow `BertEncoder` model with a vocabulary size of 100, one intermediate transformer layer, and all other parameters set to their default values. Each experiment consists of a single training loop of 50 iterations with uniformly sampled random input data of a query size of 5. The efficient squared norm functions were taken from the descriptions in Section 5 and Appendices E.1 and E.2.

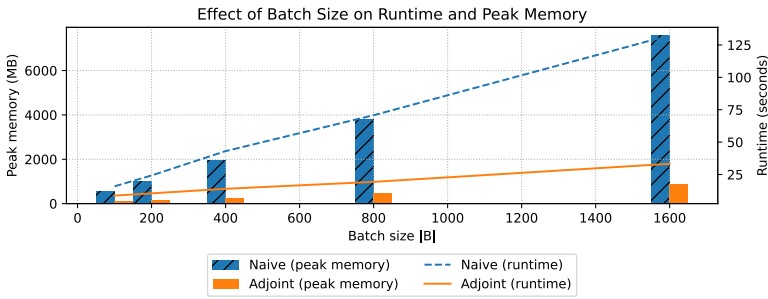

Figure 3: Runtime and memory cost graphs for training a small BERT model with batch sizes $|B| = \{100, 200, 400, 800, 1600\}$.

The results in Figure 3 demonstrate that the runtimes and peak memory usages of `Adjoint` scale better and are comparatively smaller than the corresponding ones for `Naive`.

# 7 Concluding Remarks

The analysis in Subsection 5.1 may also be applied to more complex layers whose parameter transformations $\phi_x(\cdot)$ primarily involve linear transforms. For example, it is shown in [5, Subsection 2.3] that 2D-convolution layers are equivalent to fully-connected layers when the example image is appropriately transformed to form the matrix $U_x$ in (6) and, hence, our storage and runtime savings for fully-connected layers easily apply to 2D- (and generally $n$D-) convolution layers. Another example is the multihead attention layer, whose parameter transforms are simple matrix multiplications (see Appendix E.1 for a derivation).

## 7.1 Limitations

The proposed framework only applies to layers with at least one differentiable intermediate transformation and models with differentiable losses. Moreover, the framework does not support shared trainable layers.

## Acknowledgements

The authors acknowledge the generous support from Walid Krichene (Google) and Li Zhang (Microsoft), who have meticulously reviewed the derivation and implementation of the proposed framework.

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
