## A Technical Proofs

*Proof of Proposition 4.1.* . Using the chain rule, (1), and the definitions of $\ell_x$, $\phi_x$, $\psi_x$, and $Z_x$, it follows that for any $\delta \in \mathcal{W}$ we have

$$
\begin{aligned}
D_{\bar{w}}h(x,\bar{w})[\delta] = D(\ell_x \circ \phi_x)(\bar{w})[\delta] &= D(\ell_x \circ \psi_x)(Z_x(\bar{w})) \circ DZ_x(\bar{w})[\delta] \\
&= \langle \nabla(\ell_x \circ \psi_x)(Z_x(\bar{w})), DZ_x(\bar{w})[\delta] \rangle = \langle g_x, \mathcal{A}_x \delta \rangle \\
&= \langle \mathcal{A}_x^* g_x, \delta \rangle .
\end{aligned}
$$

The conclusion follows immediately from (2) with $\psi = h(x,\cdot)$ and $w_0 = \bar{w}$. $\qquad\square$

## B Efficient Implementation of Algorithm 2

This appendix presents the technical details of efficiently implementing Algorithm 2.

### B.1 Computing Intermediate Quantities

We argue that in the setting of neural networks, Algorithm 2 can obtain the intermediate quantities

$$
\zeta_i \equiv \zeta_{i,x}, \quad g_i \equiv g_{i,x}
$$

in a **single** forward and backward (or backpropagation) pass of the given network. For ease of notation, let $i \geq 1$ be fixed and denote

$$
\Phi_i(\cdot) \equiv \phi_i(\cdot, \ldots, \phi_1(w_1, x)), \quad \tilde{\Phi}_i(\cdot) \equiv \phi_i(w_i, \cdot), \quad Z_i(\cdot) \equiv Z_{x,i}(\cdot), \quad \psi_i(\cdot) = \psi_{x,i}(\cdot),
$$

for every $i = 1, \ldots, k$ and $x \in B$.

We start with the claim about $\zeta_i$. As $Z_i(\cdot)$ and $\phi_i(\cdot, \cdot)$ are the only two functions in our setting that depend on $w_i$, it is straightforward to see that there exist functions $\alpha_i(\cdot)$ and $\beta_i(\cdot)$ such that

$$
\Phi_i(w_i) = \alpha_i \circ Z_i(w_i), \quad Z_i(w_i) = \beta_i \circ \Phi_{i-1}(w_{i-1}), \tag{9}
$$

which together yield the recursive expression

$$
\Phi_i(w_i) = \alpha_i \circ \beta_i \circ \Phi_{i-1}(w_{i-1}) \equiv \tilde{\Phi}(\Phi_{i-1}(w_{i-1})). \tag{10}
$$

Since a single forward pass obtains the values $\{\Phi_i(w_i)\}_{i=1}^k$ in sequence, it follows from (9) and (10) that we also obtain $\zeta_i = Z_i(w_i)$ as part of the forward pass.

To show the claim about $g_i$, we first notice that (9) and (10) imply

$$
\psi_i(\cdot) = \alpha_k \circ \beta_k \circ \alpha_{k-1} \circ \beta_{k-1} \circ \cdots \circ \alpha_i(\cdot)
$$

and, hence, the derivative of $\psi_i(\cdot)$ only depends on the derivatives of $\{\alpha_j\}_{j=i}^k$ and $\{\beta_j\}_{j=i+1}^k$. Since a single backpropagation step (i) obtains the derivatives (or their adjoints) of $\ell_x$, $\alpha_i$, and $\beta_i$ in the reverse order of $i = k, \ldots, 1$ and (ii) efficiently computes their sequential compositions, e.g.,

$$
D(\ell_k \circ \alpha_k \circ \beta_k \circ \cdots \circ \alpha_i)(\cdot)(\cdot),
$$

it follows that the adjoint operators $[D(\ell_x \circ \psi_i)(\cdot)]^*(\cdot)$ are also obtained from a single backpagation step. Assuming that $\zeta_i = Z_i(w_i)$ was already obtained in the first for-loop of Algorithm 2, we also obtain from these operators the gradients $g_i = \nabla(\ell_x \circ \psi_i)(\zeta_i)$.

In the next section, we give some examples of layers where the representations of the functions $\Omega_i$ can also be obtained simultaneously with the outputs $\zeta_i$.

### B.2 Computing Batch Gradients

Suppose that (5) holds for every $x \in B$ and $i = 1, \ldots, k$, and define

$$
S_i(B) := \sum_{x \in B} \ell_x \circ \psi_{i,x}(\zeta_{i,x}), \quad \hat{g}_{i,x} := \nabla_{\zeta_{i,x}} S_i(B)
$$

Let us now show that $g_{i,x} = \hat{g}_{i,x}$. First, observe that if $x, x' \in B$ and $x \neq x'$ then neither $\ell_x(\cdot)$ nor $\psi_{i,x}(\cdot)$ depend on the value of $\zeta_{i,x'}$ (and *vice versa*) for any $i = 1, \ldots, k$. Consequently,

$$\hat{g}_{i,x} = \nabla_{\zeta_{i,x}} S_i(B) = \nabla_{\zeta_{i,x}} (\ell_x \circ \psi_{i,x})(\zeta_{i,x}) + \underbrace{\sum_{x' \neq x} \nabla_{\zeta_{i,x}} (\ell_x \circ \psi_{i,x}(\zeta_{i,x}))}_{=0}$$

$$= \nabla_{\zeta_{i,x}} (\ell_x \circ \psi_{i,x})(\zeta_{i,x}) = g_{i,x}.$$

Note that since each $\zeta_{i,x}$ is computed in a single forward pass of a batch $B$ (see Appendix B.1), the quantities $\hat{g}_{i,x}$, which are gradients taken with respect to $\zeta_{i,x}$, are obtained in a single (batched) backward pass.

## C   Additional Algorithms

Algorithm 3 gives a subroutine for computing the necessary scalars used in the efficient squared norm function of the embedding layer.

---

**Algorithm 3** Computing the Nonzero Values of $n_k(x)$

---

> **Input:** $\pi(x) = [\pi_1(x), \ldots, \pi_q(x)]$
> Create an empty binary search tree $T$, where $T[k]$ denotes the value at key $k$
> **for** $i = 1, \ldots, q$ **do**
>    If $\pi_i(x)$ is not in $T$, set $T[\pi_i(x)] = 1$.
>    Otherwise, set $T[\pi_i(x)] = T[\pi_i(x)] + 1$
> **end for**
> For every key $k$ in $T$, set $n_k(x) = T[k]$.
> Return the nonzero values of $n_k(x)$.

---

## D   Decomposition Proofs

This appendix gives the derivation of the squared-normed functions $\Omega_x(\cdot)$ found in the main body of the paper.

### D.1   Fully-Connected Layer

Since $Z_x(\cdot)$ is linear, the adjoint operator of $DZ_x(V, b)(\cdot)$ is clearly given by

$$[DZ_x(V,b)]^*(g) = \begin{bmatrix} U_x^* \\ \mathcal{Q}^* \end{bmatrix} g$$

and, hence, that $\Omega_x(g) = \|U_x^* g\|^2 + \|\mathcal{Q}^* g\|^2$. It now follows from the definition of the adjoint that

$$\|U^* g\|^2 = \langle U_x^* g, U_x^* g \rangle = \langle U_x U_x^* g, g \rangle = \langle U_x U_x^*, g g^* \rangle$$

which then implies

$$\Omega_x(g) = \langle U_x U_x^*, g g^* \rangle + \|\mathcal{Q}^* g\|^2.$$

### D.2   Embedding Layer

Since $Z_x(\cdot)$ is linear, the adjoint operator of $DZ_x(W)(\cdot)$ is clearly given by

$$[DZ_x(W)]^*(g) = Y_{\pi(x)}^* g$$

and, hence, that $\Omega_i(g) = \|Y_{\pi(x)}^* g\|^2$. Now, denote $Y_\pi = Y_{\pi(x)}$, $\pi = \pi(x)$, and $\mathrm{Row}_k(M)$ to be the $k$-th row of a matrix $M$. Recall that

$$Y_\pi W = \begin{bmatrix} \mathrm{Row}_{\pi_1}(W) \\ \vdots \\ \mathrm{Row}_{\pi_q}(W) \end{bmatrix}.$$

Hence, for every $g \in \mathbb{R}^{q \times d}$ and $W \in \mathbb{R}^{r \times d}$, we have

$$\langle Y_\pi W, g \rangle = \sum_{i=1}^{q} \langle \mathrm{Row}_{\pi_i}(W), \mathrm{Row}_i(g) \rangle = \sum_{j=1}^{r} \langle \mathrm{Row}_j(W), \nu_j(g) \rangle = \langle W, Y_\pi^* g \rangle$$

where

$$\mathrm{Row}_j(Y_\pi^* g) = \nu_j(g) = \sum_{i:\pi_i=j} \mathrm{Row}_i(g).$$

Using the fact that $\mathrm{Row}_{i_1}(g) = \mathrm{Row}_{i_2}(g)$ for any $i_1$ and $i_2$ satisfying $\pi_{i_1} = \pi_{i_2} = j$, we have that

$$\mathrm{Row}_j(Y_\pi^* g) = \sum_{i:\pi_i=j} \mathrm{Row}_i(g) = n_j(x) \tilde{g}_j$$

which clearly implies $\Omega_x(g) = \|Y_\pi^* g\|^2 = \sum_{j=1}^{r} n_j(x) \tilde{g}_j$.

### D.3 Low Rank Approximation Layer

Since $Z_x(\cdot)$ is quadratic, $Z_x(\cdot)$ is Fréchet differentiable and, hence, Gateaux differentiable. Consequently, the derivative $DZ_x(V)(\cdot)$ is given by the Gateaux differential

$$\begin{aligned}
DZ_x(V)[\Delta] &= \lim_{t\downarrow 0} \frac{Z_x(V+t\Delta) - Z_x(V)}{t} \\
&= \lim_{t\downarrow 0} \frac{\langle U_x, (V+t\Delta)(V+t\Delta)^* - VV^* \rangle}{2t} \\
&= \frac{1}{2} \langle U_x, \Delta V^* + V\Delta^* \rangle + \lim_{t\downarrow 0} \frac{t}{2} \langle U_x, \Delta\Delta^* \rangle \\
&= \frac{1}{2} \langle U_x, \Delta V^* + V\Delta^* \rangle.
\end{aligned}$$

Re-arranging terms, we have that

$$DZ_x(V)[\Delta] = \frac{1}{2} \langle U_x, \Delta V^* + V\Delta \rangle = \frac{1}{2} \langle (U_x + U_x^*)V, \Delta \rangle$$

and, hence, that $[DZ_x(V)]^*(g) = g(U_x V^* + V U_x^*)/2$ for every $g \in \mathbb{R}$. Consequently, we have that

$$\Omega_x(g) = \|[DZ_x(V)]^*(g)\|^2 = \frac{g^2}{4} \|(U_x + U_x^*)V\|^2.$$

## E Additional Squared-Norm Functions

This appendix discusses decompositions of more complicated layer functions.

### E.1 Multi-Head Attention

Define the auxiliary functions $\mathrm{Softmax} : \mathbb{R}^n \mapsto \mathbb{R}^n$ and

$$\mathcal{T} : \mathbb{R}^{d_1 \times n} \times \mathbb{R}^{d_2 \times n} \times \mathbb{R}^{d_3 \times n} \times \mathbb{R}^{d_3 \times n} \mapsto \mathbb{R}^{d_3 \times n}$$

as given by

$$\left[ \mathrm{Softmax}(x) \right]_j = \frac{\exp(x_j)}{\sum_{i=1}^n \exp(x_i)},$$

$$\mathcal{T}(Q, K, V, M) = V \left[ \mathrm{Softmax}\left( \frac{K^T Q}{\sqrt{d_1}} \right) \bullet M^T \right] \in \mathbb{R}^{d_3 \times n},$$

where $\mathbf{1}$ is a vector of all ones and $A \bullet B$ denotes the Hadamard product between $A$ and $B$.

Given variables $\{W_i^Q\}_{i=1}^h \subseteq \mathbb{R}^{d_m \times d_q}$, $\{W_i^K\}_{i=1}^h \subseteq \mathbb{R}^{d_m \times d_q}$, $\{W_i^V\}_{i=1}^h \subseteq \mathbb{R}^{d_m \times d_v}$, and $\{W_i^O\}_{i=1}^h \subseteq \mathbb{R}^{d_m \times d_v}$, input queries $Q_x \in \mathbb{R}^{d_q \times n}$, input keys $K_x \in \mathbb{R}^{d_q \times n}$, input values $V_x \in \mathbb{R}^{d_v \times n}$, and mask $M_x \in \mathbb{R}^{d_v \times n}$, the multi-head attention layer function $\phi_x(\cdot)$ is given by

$$\phi_x(W) = \sum_{i=1}^h W_i^O \mathcal{T}_i(x),$$

$$\mathcal{T}_i(x) := \mathcal{T}(W_i^Q Q_x, \, W_i^K K_x, \, W_i^V V_x, \, M_x) \in \mathbb{R}^{d_v \times n}$$

where $W = (W_1, \ldots, W_h)$ and $W_i := (W_i^Q, W_i^K, W_i^V, W_i^O)$.

We now consider the squared-norm function

$$\Omega_x : \mathbb{R}^{h d_m \times n} \times \mathbb{R}^{h d_m \times n} \times \mathbb{R}^{h d_m \times n} \times \mathbb{R}^{h d_m \times n} \mapsto \mathbb{R}$$

generated by the choice of

$$Z_x(W_1, \ldots, W_h) = \begin{bmatrix} Z_x^Q(W_1^Q, \ldots, W_h^Q) \\ Z_x^K(W_1^K, \ldots, W_h^K) \\ Z_x^V(W_1^V, \ldots, W_h^V) \\ Z_x^O(W_1^O, \ldots, W_h^O) \end{bmatrix},$$

given by

$$\left[ Z_x^Q(W_1^Q, \ldots, W_h^Q) \right]_j = Z_{x,j}^Q := W_j^Q Q_x,$$

$$\left[ Z_x^K(W_1^K, \ldots, W_h^K) \right]_j = Z_{x,j}^K := W_j^K K_x,$$

$$\left[ Z_x^V(W_1^V, \ldots, W_h^V) \right]_j = Z_{x,j}^V := W_j^V V_x,$$

$$\left[ Z_x^O(W_1^O, \ldots, W_h^O) \right]_j = Z_{x,j}^O := W_j^O \mathcal{T}_j(x),$$

for $j = 1, \ldots, h$. Since each of the block functions that define $Z_x(\cdot)$ are linear, we can apply the same analysis as in the fully connected setting to obtain

$$\| [D Z_{x,j}^B]^*(g_j) \|^2 = \left\langle B B^*, g_j g_j^* \right\rangle \quad \forall B \in \{Q_x, K_x, V_x\},$$

$$\| [D Z_{x,j}^O]^*(g_j) \|^2 = \left\langle \mathcal{T}_j(x)[\mathcal{T}_j(x)]^*, g_j g_j^* \right\rangle.$$

Hence, for appropriately sized gradients $g = (g_1, \ldots, g_h)$ where $(g_j^{Q_x}, g_j^{K_x}, g_j^{V_x}, g_j^O)$, one has that

$$\Omega_x(g) = \sum_{j=1}^h \left[ \left\langle \mathcal{T}_j(x)[\mathcal{T}_j(x)]^*, (g_j^O)(g_j^O)^* \right\rangle + \sum_{B \in \{Q_x, K_x, V_x\}} \left\langle B B^*, (g_j^B)(g_j^B)^* \right\rangle \right],$$

and a representation of $\Omega_x$ may be obtained through the matrices $Q_x Q_x^*$, $K_x K_x^*$, $V_x V_x^*$, and $\{\mathcal{T}_j(x)[\mathcal{T}_j(x)]^*\}_{j=1}^h$, which only consumes $\Theta(d_q^2 + h d_v^2)$ additional storage for a batch $B$.

Notice that this is exceedingly more efficient than the naive approach of materializing each $\nabla_W h(x, W)$ for $x \in B$, which consumes $\Theta(|B|(d_m d_q + d_m d_v))$ additional storage. Moreover, the classic ghost clipping technique does not immediately apply to $\phi_x(W)$ as there is not a simple transform from $\phi_x(W)$ to some linear function of $W$.

### E.2 Layer Normalization

Given scaling variables $\gamma \in \mathbb{R}^c$, offset variables $\beta \in \mathbb{R}^c$, tolerance $\varepsilon > 0$, input $u_x \in \mathbb{R}^d$ where $c | d$, and a linear broadcasting operator $\mathcal{Q} : \mathbb{R}^c \mapsto \mathbb{R}^d$, the layer normalization layer function $\phi_x(\cdot)$ is given by

$$\phi_x(\gamma, \beta) = (\mathcal{Q}\gamma) \bullet \left[ \frac{u_x - \text{Mean}(u_x)}{\sqrt{\text{Var}(u_x) + \varepsilon}} \right] + \mathcal{Q}\beta,$$

where $A \bullet B$ denotes the Hadamard product between $A$ and $B$ and $\mathrm{Mean}(u_x)$ (resp. $\mathrm{Var}(u_x)$) is a vector in $\mathbb{R}^d$ (resp. scalar in $\mathbb{R}$) whose entries are the mean (resp. variance) of the entries in $u_x$.

We now consider the squared-norm function $\Omega_x : \mathbb{R}^c \times \mathbb{R}^c \mapsto \mathbb{R}$ generated by the choice of $Z_x(\gamma, \beta) = (\gamma, \beta)$, i.e., $Z_x(\cdot)$ is the identity function. Immediately, one has that $\Omega_x(g) = \|g\|^2$, which incurs a compute (resp. storage) cost of $\Theta(|B|c)$ (resp. $\Theta(1)$).

More interestingly, when $2c \ll d$, this approach is strictly better than both the naive approach *and* the ghost clipping technique when applied with $Z_x(\gamma, \beta) = \phi_x(\gamma, \beta)$. In the former case, it is straightforward to see that we incur a compute (resp. storage) cost of $\Theta(|B|c)$ (resp. $\Theta(|B|c)$). To analyze the latter case, let $D_x \in \mathbb{R}^{d \times d}$ be a diagonal matrix given by

$$[D_x]_{ii} = \left[ \frac{u_x - \mathrm{Mean}(u_x)}{\sqrt{\mathrm{Var}(u_x) + \varepsilon}} \right]_i \quad i = 1, \dots, d,$$

and observe that

$$\phi_x(\gamma, \beta) = \underbrace{\left[ \begin{array}{cc} D_x \mathcal{Q} & \mathcal{Q} \end{array} \right]}_{=:U_x} \left[ \begin{array}{c} \gamma \\ \beta \end{array} \right].$$

Consequently, the classic ghost clipping technique yields the decomposition

$$\Omega_x(g) = \langle U_x U_x^*, gg^* \rangle = \langle D_x \mathcal{Q}\mathcal{Q}^* D_x + \mathcal{Q}\mathcal{Q}^*, gg^* \rangle$$

for $g \in \mathbb{R}^d$, which is incurs a steep compute (resp. storage) cost of $\Theta(d^2 c + d^2 |B|)$ (resp. $\Theta(d^2)$) for general $\mathcal{Q}$ and $2c \ll d$.

# F  Additional Experiments

## F.1  Effect of Batch Size on Fully-Connected Layers

Figure 4 presents numerical results for the same set of experiments as in Subsection 5.1 but for different batch sizes $|B|$ instead of the output dimension $q$.

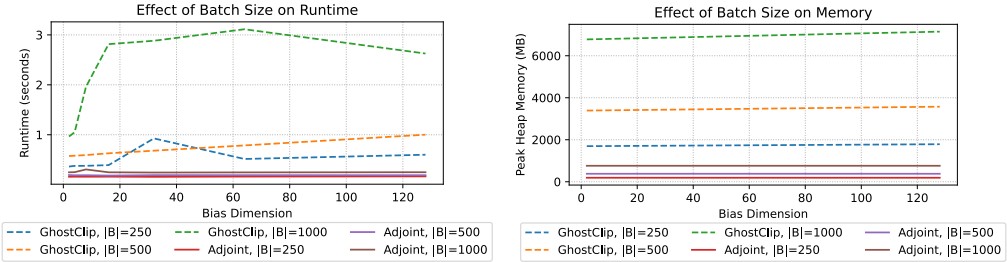

Figure 4: Runtime and memory cost graphs for fully-connected layer computations with bias dimensions $m = \{2^1, 2^2, \dots, 2^{11}\}$ and batch sizes $|B| = 250, 500, 1000$.

Similar to Subsection 5.1, the results in Figure 4 are more favorable towards `Adjoint` compared to `GhostClip`.