# OpenReview forum: "A Unified Fast Gradient Clipping Framework for DP-SGD"
_NeurIPS.cc/2023/Conference — NeurIPS 2023 poster_

### Official Review · Reviewer_6FUB · 2023-06-30

**Soundness:** 3 good
**Presentation:** 3 good
**Contribution:** 3 good
**Rating:** 6
**Confidence:** 1

**Summary:**

The paper provides a unification of ad-hoc analysis and interpretations of the ghost clipping algorithm under a single framework. It also shows that for certain operations, such as fully-connected and embedding layer computations, further improvements to the runtime and storage costs of existing decompositions can be deduced using certain components of the proposed framework.

**Strengths:**

- **Originality**: The paper proposes a novel framework for fast gradient clipping used in DP-SGD. This is a significant theoretical contribution to the field of privacy of machine learning.
- **Quality**: The paper is well-organized. A thorough explanation of the proposed framework, including mathematical derivations and experimental results is provided.
- **Clarity**: The paper is easy to follow and understand, even for readers who are not experts in the field. The authors provide clear explanations of the background, motivations, concepts, and techniques used, and the experimental results are presented in a concise and understandable manner.
- **Significance**: The paper has significant implications for the specific field of privacy of machine learning. It provides a unified perspective on the ghost clipping algorithm in DP-SGD.

**Weaknesses:**

The paper focuses only on a specific technique of gradient clipping, i.e., ghost clipping. Whether this ghost clipping lies at the heart of the big field of privacy of deep learning or not is unclear to me.

**Questions:**

How can the proposed framework be generalized beyond the ghost-clipping algorithm?

**Limitations:**

Limitations are not discussed by the authors. Perhaps the possible relation or extension to other clipping algorithms can be mentioned in the final as a separate concluding section which is now missing.

---

> ### Author Rebuttal · Authors · 2023-08-07
>
> Thank you for taking the time to review our manuscript. Below, you find some responses to some of your questions and concerns.
>
> > The paper focuses only on a specific technique of gradient clipping, i.e., ghost clipping. Whether this ghost clipping lies at the heart of the big field of privacy of deep learning or not is unclear to me.
>
> As far as we are aware, there are no other techniques in the literature for quickly clipping gradients in neural networks aside from ghost clipping.
>
> > How can the proposed framework be generalized beyond the ghost-clipping algorithm?
>
> I am unsure what the reviewer is asking here. The ghost-clipping technique is a special instance of our proposed framework, and not the other way around.

---

> > ### Comment · Reviewer_6FUB · 2023-08-19
> >
> > Thank you very much for your clarification. You may ignore my question in the "Questions" because I have misunderstood some perspectives. I keep my score as "Weak Accept" after reading other reviewers' comments.

---

### Official Review · Reviewer_zBGn · 2023-07-03

**Soundness:** 4 excellent
**Presentation:** 4 excellent
**Contribution:** 3 good
**Rating:** 7
**Confidence:** 3

**Summary:**

This paper provides a unified framework for efficiently computing the gradient norms of individual examples for a wide range of neural network architectures, which significantly decreases runtime and storage costs for implementing DP-SGD.

**Strengths:**

1. The considered problem is important: Computing per-example gradients is a computation bottleneck for previous implementations of DP-SGD.
2. Theoretical and empirical improvements over baselines are clearly presented and significant. Tables 1 and 2 summarize the time and storage complexity for previous work and the proposed algorithm, and Figures 1 and 2 summarize the observed improvements in experiments. The theoretical and the empirical improvements are both convincing.
3. The presentation is rigorous and clear.

**Weaknesses:**

1. The experimental evaluation does not provide a big-picture idea of the end-to-end savings provided by the proposed framework. Figures 1 and 2 exhibit the runtime/storage costs only for the gradient computation operation, and it does not show how the overall costs are affected. If gradient computation accounts for a small portion of the overall runtime, then the overall time savings will not be significant. As it stands, the experimental results do not give a concrete idea for the overall savings in runtime, which is the metric of interest for practitioners.
2. It would be nice to see an empirical evaluation for the more popular layer types mentioned in the text, e.g. convolution and attention.

**Questions:**

1. What is the overall reduction in runtime/storage cost provided by the proposed framework?
2. How does the empirical performance of the proposed framework compare to that of ghost clipping for convolution layers and attention layers?

**Limitations:**

There is no discussion of societal impact beyond the underlying (implied) importance of user privacy. However, I don't think any further discussion of societal impact is necessary for this work.

---

> ### Author Rebuttal · Authors · 2023-08-07
>
> Thank you for your positive evaluation of our work. We hope the below comments will address the issues you brought forth.
>
> > The experimental evaluation does not provide a big-picture idea of the end-to-end savings provided by the proposed framework. Figures 1 and 2 exhibit the runtime/storage costs only for the gradient computation operation, and it does not show how the overall costs are affected. If gradient computation accounts for a small portion of the overall runtime, then the overall time savings will not be significant. As it stands, the experimental results do not give a concrete idea for the overall savings in runtime, which is the metric of interest for practitioners.
>
> For feed forward neural networks the forward and backward steps use a similar amount of compute and memory, thus highlighting the results of decreased gradient computation time seemed reasonable to us. However, in order to provide a more complete picture we can add end-to-end runtime analysis of our technique compared to the naive clipping technique.
>
> > It would be nice to see an empirical evaluation for the more popular layer types mentioned in the text, e.g. convolution and attention.
>
> As the convolution operator is equivalent to a fully-connected layer (see [1] below) and the parameter transforms of multihead attention layers are merely matrix multiplications (see our Appendix E.1), we expect their empirical performance to be similar to what is shown for fully connected layers (see Figures 1 and 2).
>
> [1] Bu, Z., Mao, J., & Xu, S. (2022). Scalable and efficient training of large convolutional neural networks with differential privacy. Advances in Neural Information Processing Systems, 35, 38305-38318.
>
> > What is the overall reduction in runtime/storage cost provided by the proposed framework?
>
> In our experiments, the large majority of the work performed in the algorithm was in the computation of the gradient norms. Hence, the graphs in Figures 1 and 2 are also indicative of the overall cost reduction provided by our framework.
>
> > How does the empirical performance of the proposed framework compare to that of ghost clipping for convolution layers and attention layers?
>
> See our response above about convolution and attention layers.

---

> > ### Comment · Reviewer_zBGn · 2023-08-18
> >
> > Thank you for your response. I agree with you that the time taken for gradient computation is likely to be representative of the overall computation time, but the accuracy of this representation is not certain until it is actually measured. It would still be nice to see a comparison of the overall running time. I feel similarly about a comparison for the other layer types. It seems likely that the other layer types will behave similarly as the fully connected layer, but we cannot know for sure until it is measured. Still, these are small details and I am satisfied with the quality of the paper, so I will maintain by recommendation for acceptance.

---

### Official Review · Reviewer_mrhy · 2023-07-04

**Soundness:** 3 good
**Presentation:** 3 good
**Contribution:** 3 good
**Rating:** 7
**Confidence:** 3

**Summary:**

This paper unifies a framework that enables us to apply ghost clipping algorithms to new layers of neural networks, apart from fully connected, feed-forward layers. In particular, the paper discusses efficient implementations of computing norms for running DP-SGD to train neural networks.

**Strengths:**

The contributions of this paper are as follows:

1. It shows how to utilize the proposed framework to apply ghost clipping techniques to new layers for DP-SGD on training neural networks. Three main examples are included to discuss how to implement efficient algorithms to compute the gradient norm and to run DP-SGD.

2. Experimental results evaluate the effectiveness of their proposed framework over the classical ghost clipping framework in run time and memory requirements.

**Weaknesses:**

Literature on the importance of DP-SGD and clipping operators is lacking in this paper. It is better to include them in Section 3 (Previous work) or in the introduction section to motivate the study of this paper in a broader impact.

**Questions:**

N/A

---

> ### Author Rebuttal · Authors · 2023-08-07
>
> Thank you for your positive impression of our work! Below you will find our comments addressing your main concern.
>
> > Literature on the importance of DP-SGD and clipping operators is lacking in this paper. It is better to include them in Section 3 (Previous work) or in the introduction section to motivate the study of this paper in a broader impact.
>
> In the second paragraph of the “Introduction” section (Section 1), we discussed the importance of DP-SGD, motivated the use of gradient clipping, and remarked on the importance of private training. Moreover, in the third paragraph of the “Introduction” section and the “Previous work” section (Section 3), we discussed why the process of clipping is hard to scale to large models. If you have any specific pointers related to the importance of DP-SGD we would be happy to add them as additional references.

---

### Official Review · Reviewer_PneF · 2023-07-05

**Soundness:** 2 fair
**Presentation:** 2 fair
**Contribution:** 3 good
**Rating:** 5
**Confidence:** 3

**Summary:**

The paper studies the memory/time complexity of the per-sample gradients norm computation step in the DP-SGD algorithm. The authors present a general theoretical framework that generalizes the idea of ghost norm computation (computing the norm of the persample gradients without having to store the gradients), and can be applied to arbitrary layers of a DNN (if an ad hoc decomposition can be found). The authors show early experiments on the benefits of their methods compared to previous ghost clipping techniques.

**Strengths:**

originality: - Computing persamble gradient norms is a real bottleneck when training DNNs with DP-SGD. Typically, the Opacus library (Pytorch for privacy) stores the gradients per sample before computing the norms, which can be memory intensive, especially for large architectures. Other methods have been proposed to compute the norm without storing the gradients (ghost norm), but no general framework has been proposed yet. This paper gives a clear problem formulation and a general theoretical framework to approach this problem. I was very happy to see a theoretical work that tries to tackles this issue.
quality: The math are correct
clarity:  ok
significance: using the theory of linear operator to improve time and space complexity of DP-SGD is good novelty that could benefit the community

**Weaknesses:**

- Typos: l108 “authors”
- Overall, the experiments are very insufficient. They are performed with batches of size 1, for which the vanilla per-sample gradient storage method has a better running time, but with which it is never compared. The authors should investigate in detail how their method would work with GPUs to see how it would be beneficial in practice: the complexity bounds given are linear in the batch size, but in practice we benefit greatly from parallel computation/optimized matrix multiplication.
- Experiments should be performed on whole (even small) networks used in practice, with batches of size greater than one. In Li et al (2022), they compare ghost norm to opacus when training a GPT2-like architecture and find similar runtimes; a similar comparison could be done here (even with much smaller versions of GPT2 when computational power is limited).
- It is not the same abstract in OpenReview and in the article. The first states that savings can be as high as 150x: this could be the case in very specific situations chosen for the experiments, but the authors do not elaborate on why these choices are realistic in practice.
- The section on related work is sparse, with very few papers cited (no mention of why gradients need to be clipped, no mention of the whole deep learning literature where this is a real bottleneck (Opacus))
- No code for reproduction


**Questions:**

- Typically Opacus (storing per sample gradients) or ghost norm will be 2x slower than non DP training. How can the proposed method be 150x faster? That would mean that the proposed method is faster than non DP training?
- Why does the green curve start decreasing at some point in the memory figure for the embedding layer?
- For the embedding layer, the authors state that q=1 in practice but show memory/runtime for q = {1000, 2000, …, 10000}. Can you please clarify on these choices for comparison?
- The paper states that it can be applied to arbitrary layers, but the examples are linear layers (which were already tackled in ghost norm previous work). What are the conditions for the existence of the decomposition in proposition 4.1? What do the authors have in mind by “efficient manner” (line 156)? When is that possible?


**Limitations:**

The paper does not include a limitation section.
Suggestions for improvement would be to perform a rigorous empirical study of the proposed method, with entire architectures and using micro batches, and see if can easily used with an existing DL library (pytorch, JAX...).

---

> ### Author Rebuttal · Authors · 2023-08-07
>
> Thank you for your insightful comments towards improving the quality of our manuscript. We hope that the comments and considerations below will help improve your overall opinion of our work.
>
> > Typos: l108 “authors”
>
> This will be fixed in the revision.
>
> > Overall, the experiments are very insufficient...
>
> We will add additional experiments that vary the batch size as well as compare GPU costs against CPU costs. As a preview of the results, here are the fully-connected memory and runtime profiles of ghost clipping (GC) vs. our method (A) for varying bias dimensions $m$ and batch sizes $|B|$.
>
> **Runtimes (in seconds)**
>
> | m | GC (\|B\|=250) | A (\|B\|=250) | GC (\|B\|=500) | A (\|B\|=500) | GC (\|B\|=1000) | A (\|B\|=1000) |
> |-|-|-|-|-|-|-|
> | 16 | 0.224 | 0.061 | 0.369 | 0.094 | 0.865 | 0.154 |
> | 32 | 0.230 | 0.064 | 0.370 | 0.089 | 0.907 | 0.156 |
> | 64 | 0.239 | 0.063 | 0.430 | 0.092 | 0.850 | 0.154 |
> | 128 | 0.285 | 0.061 | 0.562 | 0.090 | 1.354 | 0.156 |
>
> **Peak Memory (in MB)**
>
> | m | GC (\|B\|=250) | A (\|B\|=250) | GC (\|B\|=500) | A (\|B\|=500) | GC (\|B\|=1000) | A (\|B\|=1000) |
> |-|-|-|-|-|-|-|
> | 16 | 1705 | 190 | 3410 | 380 |7569 | 761 |
> | 32 | 1717 | 190 | 3433 | 380 | 7616 | 761 |
> | 64 | 1740 | 190 | 3480 | 380 | 7710 | 761 |
> | 128 | 1787 | 190 | 3574 | 380 | 7898 | 761 |
>
> > Experiments should be performed on whole (even small) networks ...
>
> We will add additional experiments on a small BERT model to see how our framework performs in practice. We do want to point out that Li et al (2022) show that some models couldn’t be trained with Opacus simply because the per example clipping made GPUs run out of memory. We expect our results to be very similar in nature.
>
> > ...the authors do not elaborate on why these choices are realistic in practice...
>
> The numerical experiments in Section 6 were specifically chosen to demonstrate the better scalability of our implementation compared to ghost clipping and, hence, the parameter choices may not be typically observed in other literature.
>
> We will add additional end-to-end experiments that utilize more practical parameter choices.
>
> > The section on related work is sparse...
>
> The importance of clipping gradients is discussed in the second paragraph of the “Introduction” section (Section 1). Moreover, the steps in Algorithm 1 clearly show that DP-SGD cannot be implemented without gradient clipping (hence, its importance).
>
> We will add additional references to works where the clipping step is a serious bottleneck to training DP models. Also, we would be happy to consider specific works the referee has in mind.
>
> > No code for reproduction
>
> Code will be made publicly available if the paper is accepted.
>
> > Typically Opacus (storing per sample gradients) ... will be 2x slower than non DP training...
>
> As far as we are aware, there do not appear to be any studies that compare Opacus / ghost norm / non-DP gradient norm computations on individual layers. Hence, the “2x” slower performance is not informative of the layer-specific efficiency of ghost norm / Opacus vs non-DP for different choices of layer parameters.
>
> Consequently, the 150x improvement of our framework is due to the large layer (or query) dimension sizes that are tested in our experiments. A simple analogy (with rough guesses) is that BubbleSort is only 2x slower than QuickSort when the number of elements $n$ is less than 20, but can be more than 1000x slower when $n$ is greater than 5000.
>
> In view of the above arguments, we do not agree that our experiments imply our proposed framework is faster than non-DP training. In fact, our framework is strictly slower because it performs one more backward and forward pass of the model compared to non-DP training (see lines 117, 137, 140, 169, and 175 in the original submission for references of this).
>
> > Why does the green curve start decreasing at some point in the memory figure for the embedding layer?
>
> We believe it is due to the randomness of the generated embedding indices.
>
> > For the embedding layer, the authors state that q=1 in practice...
>
> For added clarification, it is stated (in Subsection 5.2) that a commonly used case is q=1. In practice, models may also use embedding layers with q >> 1. For instance, transformer models require q to be the size of the sentence and personalization  models, such as those used in online advertising, may encode arbitrary information about a webpage’s context which can result in multiple queries to a model.
>
> > The paper states that it can be applied to arbitrary layers, but the examples are linear layers...
>
> We respectfully disagree about the first statement. Subsection 5.3 gives the analysis of a nonlinear layer, and the analyses on ghost clipping from prior works did not consider the sparsity of the related linear operators (this is why we are able to make a drastic improvement for embedding layers in particular).
>
> In general, the decomposition in Proposition 4.1 is feasible when $\phi_x$ can be decomposed into the composition of at least two Fréchet differentiable functions.
>
> Examples are given in Section 5 about what “efficient manner” means (on line 156). In short, this is the case when the operator $A^*$ in Proposition 4.1 admits a low-rank decomposition or is sparse. When exactly this is the case will depend on the characteristics of the layer function $\phi_x$ (in Proposition 4.1) and the chosen decomposition $\phi_x = \psi_x \circ Z_x$ (see the examples in Section 5 and Appendix E for some details).
>
> > The paper does not include a limitation section...
>
> We will add a section discussing the limitations of our framework. Moreover, we will add experiments that compare our method on more complex model architectures.

---

> > ### Comment · Reviewer_PneF · 2023-08-18
> > **Thanks**
> >
> > Thank you very much for your detailed response.
> >
> > - It would indeed be very insightful to get (empirical) results of runtime/memory for a network + HP commonly used in practice.
> >
> > - There is also the recent work of Bu et al (ICML 2023): Differentially Private Optimization on Large Model at Small cost, which is also be improving on the original Ghost norm. It could be interesting to compare to them too, what do you think?
> >
> > - "We will add additional references to works where the clipping step is a serious bottleneck to training DP models. Also, we would be happy to consider specific works the referee has in mind"
> > Training with DP-SGD is more computationally intensive than non private training for two main reasons: the first one is gradient clipping, which the current paper is tackling, and the other reason is that to attain good privacy utility tradeoff, it is better to use "mega batches": see De et al (2022): Unlocking High-Accuracy Differentially Private Image Classification through Scale, or Sander et al (2023): TAN Without a Burn: Scaling Laws of DP-SGD, or Yu et al (2023): ViP: A Differentially Private Foundation Model for Computer Vision, and the ghost norm paper that you are already citing, to cite only a few.
> >
> > Overall compute is one of the main bottlenecks to improve the performance of large architectures trained with DP-SGD , which puts your work as possibly very impactful. I am raising my score to 5.

---

> > > ### Author Response · Authors · 2023-08-19
> > >
> > > Thank you for the feedback and additional references! We will definitely try to incorporate them in the revision.
> > >
> > > >  There is also the recent work of Bu et al (ICML 2023): Differentially Private Optimization on Large Model at Small cost, which is also be improving on the original Ghost norm. It could be interesting to compare to them too, what do you think?
> > >
> > > If we understand the work of (Bu et. al., 2023) correctly (specifically Appendix D.2), the improvement over Ghost Clipping seems to be from removing the need to store/compute certain extraneous parameter gradients during the backward pass of DP-SGD. While our implementation already does this removal in theory (see how the steps in Algorithm 2 on page 5 are described) and in our numerical experiments, we will (i) make this fact explicitly known in the revision and (ii) duly cite the work of (Bu et. al., 2023)

---

### Official Review · Reviewer_zMyL · 2023-07-07

**Soundness:** 3 good
**Presentation:** 3 good
**Contribution:** 3 good
**Rating:** 6
**Confidence:** 3

**Summary:**

This work unifies the analysis of the ghost clipping algorithm, which was previously analyzed for specific architectures only. The proposed framework extends previous results to a wide class of network architectures including new applications and provides asymptotically faster computation times than the previous case-by-case approaches. Some numerical simulations confirm the effectiveness of the proposed algorithm demonstrating faster computation time in practice.

**Strengths:**

The paper is nicely structured and well motivated. The considered family of functions (NN architectures) is quite general and the proposed approach introduced by Proposition 4.1 seems to be very principled. Despite of the generality, the strategy proposed in Proposition 4.1 can be implemented for a number of important special classes of functions such as fully-connected networks, embedding layers and low rank approximation layers. The authors demonstrate improved runtime and storage costs for this algorithm in these three special cases.


**Weaknesses:**

-- The main motivation of the work is that when the batch sizes of DP-SGD is large, the runtime and storage can be large. Why this problem cannot be simply solved by using parallel computing (assigning different batches to different workers)? If this is one possible approach, I believe this should be mentioned and discussed in the related work.

-- It seems that the assumptions on the loss function and the layers $\phi_x$ are not explicitly stated. It is unclear to me if it is sufficient to have all $\phi$ and the loss function to be Frechet differentiable or any additional technical assumptions are needed for Proposition 4.1. Why the sub gradient of $Z_x$ might not be unique while $\nabla h$ and $g$ in this proposition are unique?

**Questions:**

- Why in Tables 1 and 2 some entries come with $\Theta(\cdot)$ and some $O(\cdot)$? Are there lower bounds for storage and runtime costs for these settings?

- Does equation (3) hold for the last layer only?

- Typo in line 54.


---
Update: The authors addressed my concerns and I would like to increase my score from 5 to 6 to further support the acceptance.

**Limitations:**

-

---

> ### Author Rebuttal · Authors · 2023-08-07
>
> Thank you for taking the time to thoroughly review our manuscript. We hope that our responses below will help improve your impression of our work.
>
> > The main motivation of the work is that when the batch sizes of DP-SGD is large, the runtime and storage can be large. Why this problem cannot be simply solved by using parallel computing (assigning different batches to different workers)? If this is one possible approach, I believe this should be mentioned and discussed in the related work.
>
> In short, parallel computing does not solve the scalability issues of naive clipping compared to our approach, especially when certain dimensions of the problem (aside from batch size) are large (on the order of thousands or tens of thousands). This issue is explicitly made clear in Tables 1 and 2, where our proposed framework scales much better with other problem constants (e.g., $r$, $q$, $p$ and $d$) compared to the naive and ghost clipping approaches.
>
> Also note that all three approaches considered in Tables 1 and 2 will (asymptotically) have the same benefits from switching to a parallel computing framework. Specifically, if batches were split among $s$ workers in parallel, the constant $|B|$ in Tables 1 and 2 would change to $|B|/s$ for all algorithms.
>
> > It seems that the assumptions on the loss function and the layers $\phi_x$ are not explicitly stated. It is unclear to me if it is sufficient to have all $\phi$ and the loss function to be Frechet differentiable or any additional technical assumptions are needed for Proposition 4.1. Why the sub gradient of $Z_x$ might not be unique while $\nabla h$ and $g$ in this proposition are unique?
>
> To be more precise, we will add some explicit assumptions just before Subsection 4.1 and make the differentiability of the key functions in Proposition 4.1 more clear. Specifically, we will make the explicit assumption that
>
> * $\ell(x,\cdot)$, $\phi_1(\cdot, x)$, \ldots, $\phi_k(\cdot, x)$ are Fréchet differentiable;
> * each function $\phi_i(\cdot, x)$ can be decomposed into the composition of at least two Fréchet differentiable functions;
> * the functions $\ell_x$, $\phi_x$, $\psi_x$, and $Z_x$ are Fréchet differentiable.
>
> In the revision, we will assume $Z_x$ is differentiable so that its (sub)gradient is unique.
>
> As an aside, note that the choice of $Z_x$ itself is not unique. Indeed, if $\phi_x = \psi_x \circ Z_x$ then for any $\alpha > 0$ the functions $Z_x’=\alpha Z_x$ and $\psi_x’ = \psi_x / \alpha$ also satisfy $\phi_x = \psi_x’ \circ Z_x’$.
>
> > Why in Tables 1 and 2 some entries come with $\Theta(\cdot)$  and some $O(\cdot)$? Are there lower bounds for storage and runtime costs for these settings?
>
> $\Theta(\cdot)$ is used for computations where we know the exact (up to universal constants) amount of storage/runtime needed to compute gradient norms with respect to certain dimensions. $O(\cdot)$ is used when the storage/runtime may depend on other features of the problem that are not dimensions (e.g., sparsity of the queries for embeddings layers). For these cases, an upper bound is more appropriate.
>
> As far as we know, there are no informative lower bounds for computing gradient norms. Naively, though, there is the $\Omega(|B|)$ lower for storing/computing the scalars corresponding to each example’s gradient norm in a batch.
>
> > Does equation (3) hold for the last layer only?
>
> Relation (3) may hold for any layer in a neural network. However, the functions $\phi_x$ between layers may differ. Note that this is why the corresponding $\phi_x$ functions in Algorithm 2, specifically $\{\psi_{x,i} \circ Z_{x,i}\}$, are indexed by the layer index $i$.
>
> > Typo in line 54.
>
> This will be fixed in the revision.

---

### Decision · Program_Chairs · 2023-09-21

**Decision:**

Accept (poster)

**Comment:**

The paper performs a synthesis of ad hoc analyses and interpretations of ghost clipping algorithms, bringing them together in a single, comprehensive framework.
This family of methods aim at solving the computational bottleneck in DP stemming from the computation of per-sample gradient norms. Furthermore, the authors establish that, for particular architectures, the proposed framework enhances the efficiency of existing decompositions, yielding improvements in terms of execution time and storage.

The reviewers and I concur that the paper is an interesting contribution and should be accepted.

Please take into consideration reviewers' comments before submitting the final version of your manuscript.